# Evaluation of the Effect of Duration on Retinal Nerve Fiber Layer Thickness and Choroidal Thickness in Exfoliation Syndrome and Exfoliative Glaucoma

**DOI:** 10.3390/diagnostics13020314

**Published:** 2023-01-14

**Authors:** Müslüm Toptan, Ali Simsek

**Affiliations:** Department of Ophthalmology, Faculty of Medicine, Harran University, Sanliurfa 63200, Turkey

**Keywords:** choroid, exfoliative glaucoma, exfoliation syndrome, retinal nerve fiber layer, optical coherence tomography

## Abstract

Aims: To compare retinal nerve fiber layer (RNFL) thickness and choroidal thickness in patients with exfoliation syndrome (XFS) and exfoliative glaucoma (XFG) for 1–5 years or 6–10 years compared to healthy volunteers. Methods and Material: Seventy-eight eyes of 78 patients with XFG, 78 eyes of 78 patients with XFS, and 83 eyes of 83 healthy individuals were included in this prospective study. SD-OCT data for choroid thickness and RNFL were recorded. Results: RNFL thickness was statistically significantly lower in eyes with XFG and XFS than in the control group (*p* < 0.001). Macular choroidal thickness decreased significantly in the XFG group compared to the XFS and control groups (*p* < 0.001). No significant difference was observed between the XFS and control groups (*p* > 0.05). In terms of choroidal and RNFL thicknesses by years in XFG and XFS patients, values were lower in the patients diagnosed 6–10 years previously than in those diagnosed 1–5 years previously. However, the difference was not statistically significant (*p* > 0.05). Conclusions: Thinning of both choroidal and RNLF thickness in XFG and XFS patients may mean that PEX material is an important risk factor for the progression of XFS to XFG. In addition, thinner choroid and RNLF thickness in the 6–10 years groups show the effect of PEX material and the importance of time.

## 1. Introduction

Exfoliation syndrome (XFS) is a systemic disease characterized by progressive production and deposition of the fibrillar material known as pseudoexfoliation (PEX), generally seen in the anterior segment and the pupils, and in extraocular tissues including the skin [1]. Exfoliative glaucoma (XFG) develops as a result of obstruction of the trabecular meshwork pattern by PEX material and pigment is the most common form of secondary open-angle glaucoma. XFG represents 25% of open-angle glaucoma cases and is a leading identifiable cause of such glaucoma [2].

Elevation and fluctuations may be seen in intra-ocular pressure (IOP) in patients with XFS. This can thus lead to XFG, which exhibits more rapid progression compared to primary open-angle glaucoma (PAAG) [3]. At a certain level of IOP, there is a greater possibility of damage to the optic nerve in patients with XFS, and the presence of PEX material has been identified as the most important independent risk factor for glaucoma progression [4]. In addition, optical coherence tomography (OCT) studies have reported early structural changes in RNFL thickness in XFS [5].

PEX material can affect structures in the anterior segment, such as the corneal endothelium, the anterior lens capsule, and the trabecular meshwork, and has also been shown to be capable of affecting structures in the posterior segments, such as the posterior ciliary arteries, the vortex veins, and the central retinal veins [6]. XFS has been linked to decreased ocular blood flow [7]. This decreased ocular blood flow manifests as a decrease in choroidal venous calibration. Recent advances in OCT have made it possible to measure choroidal thickness in vivo. However, the relationship between choroidal blood flow and choroidal thickness is still unclear [8]. While studies have investigated this, the effects of PEX material on choroidal blood flow and choroidal thickness remain controversial [7,9,10]. Considering the role of the choroidal vasculature in blood flow in the laminar and prelaminar regions of the optic nerve head, the choroid may be an appropriate target for investigation in glaucomatous patients [11].

The purpose of the present study was to compare RNFL and choroidal thickness in patients with XFS and XFG with healthy volunteers, and to determine the effects of exposure time to PEX material.

## 2. Material and Methods

Seventy-eight eyes of 78 patients with XFG, 78 eyes of 78 patients with XFS, and 83 eyes of 83 healthy individuals presenting to the Harran University Medical Faculty Eye Diseases Department, Turkey, between August 2018 and January 2020 were included in this retrospective study. Approval for the study was granted by the Harran University Institutional Evaluation Committee and Ethical Committee (10/08/2018-E.31565) and was conducted in line with the principles of the Declaration of Helsinki. All participants gave signed, written consent before the study began.

All cases included in the study underwent detailed ophthalmological examinations including best corrected visual acuity (BCVA), refraction, biomicroscopic examination, IOP measurement using Goldmann applanation tonometry, gonioscopy, and dilated fundus examinations. Axial length and Central Corneal Thickness (CCT) were measured using an optic biometer (Lenstar LS 900; Haag Streit AG, Koeniz, Switzerland), and visual field examination was performed with 24-2 software and a Humphrey perimeter (Carl Zeiss Meditec, Dublin, CA, USA). The XFS group consisted of patients with PEX material deposition in the pupillary margin and/or lens capsule, with IOP below 21 mmHg without medication, and with normal optic disc and normal visual field findings (no localized defects or scotoma). The XFG group consisted of patients with PEX material in the anterior lens capsule, the pupillary margin and/or the anterior chamber angle at gonioscopy. XFG group Drug-free IOP > 21 mmHg and with glaucomatous optic nerve damage (especially thinning or notching in the superior and/or inferior quadrant), and loss of visual field (a cluster of points with sensitivity loss, with at least one at the *p* < 0.1 level on the pattern deviation map). Glaucoma stage was determined by using the visual field MD values. Patients with an MD value ≥ −6 dB were considered early glaucoma, −6 dB to −12 dB as middle-stage glaucoma and ≤−12 dB as advanced glaucoma [12]. Early glaucoma group was included in our study. None of our cases with PEG had undergone glaucoma surgery.

The control group was selected from individuals with IOP below 21 mmHg and with no optic nerve damage, and no systemic disease or medication use. Individuals aged under 18, with BCVA less than 20/40, with chronic systemic diseases such as arterial hypertension or diabetes mellitus, with histories of intraocular surgery, or with retinal diseases such as macular degeneration, diabetic retinopathy of inflammatory eye disease were excluded from the study. Myopic and hypermetropic patients and individuals with astigmatic refractive error greater than ±1.0 D were also excluded. In addition, patients using prostaglandin group drugs were excluded from the study.

The RNFL and choroid were imaged with undilated pupils using spectral domain OCT (Spectralis, Heidelberg Engineering, Heidelberg, Germany). Quality control for all scans was performed before enrollment. To improve the visualization of the choroid, the instrument’s enhanced depth imaging mode was used in combination with automatic real-time eye tracking and frame averaging. Each subject was imaged by both ophthalmologists, and the two values, captured by two different ophthalmologists, were averaged for analysis. All differences between measurements taken by the blinded ophthalmologists for each subject were within 10% of the mean value. All examinations were performed between 09:00 and 12:00 a.m. to exclude diurnal variations [13]. Right eye values were used for statistical analyses. Retinal nerve fiber layer thickness was determined from the optic nerve head scan. A volumetric scanning protocol was used, imaging a 15 by 15 region surrounding the optic nerve head (circle scan size 3.4 mm). Choroidal thickness was measured manually by marking the distance between the outer margin of the retinal pigment epithelium and the choroid-sclera junction. In addition to subfoveal choroidal thickness, choroidal thickness was also measured 1.500–3.000 µm nasal to the fovea and 1.500–3.000 µm temporal to the fovea (Figure 1).

Peripapillary RNFL thickness parameters were calculated automatically using fast RNFL scan mode. The software produced a thickness profile in the temporal-superior-nasal-inferior fields with standard 12-degree circular scanning. The software also calculated mean thickness values (μm) for the global optic disc, and each of the six sectors focused on the disc (temporal, temporal superior, temporal inferior, nasal, nasal inferior, and nasal superior) (Figure 2).

Patients presenting to the clinic who had been followed-up for five years or less since diagnosed constituted the 1–5-year group, and those who had been followed up for six years or more constituted the 6–10-year group.

## 3. Statistical Analysis

Statistical analyses were performed on the SPSS 23.0 software (IBM SPSS Inc., Chicago, IL, USA). The Shapiro Wilk test was applied to assess normality of data distribution. Normally distributed variables were expressed as mean ± standard deviation and non-normally distributed data as median (interquartile range). Categorical variables were expressed as number (n) and percentage (%). Since the data were not normally distributed, the Kruskal-Wallis test was employed for three-way comparisons, and the Mann Whitney U test for two-way comparisons. Pearson’s Chi-square test was used to compare categorical variables. The confidence interval (CI) was set at 95%, and *p* values < 0.05 were regarded as statistically significant.

## 4. Results

The demographic characteristics and clinical findings of the individuals in all groups are shown in Table 1. No significant difference was determined among the three groups in terms of age, sex, IOP, axial length, or corneal thickness values (*p* > 0.05). Comparison of the XFG, XFS, and control groups in terms of mean deviation (MD) and pattern standard deviation (PSD) values revealed a statistically significant difference in the XFG group (*p* < 0.001).

Statistically significant differences in RNFL thickness were observed between the groups. RNFL thickness was significantly lower in all quadrants in eyes with XFG compared to the XFS and control groups (*p* < 0.001). Significant differences were detected between the XFS and control groups in temporal, inferiotemporal, and mean values (*p* < 0.001) (Table 2).

Macular choroidal thickness was significantly lower in the XFG group compared to the XFS and control groups (*p* < 0.001). No significant difference was detected between the XFS and control groups (*p* > 0.05) (Table 3).

When duration disease in patients with XFG and XFS was classified as 1–5 years or 6–10 years, choroidal and RNFL thicknesses were lower in the 6–10-year group among the XFG and XFS patients. However, the difference was not statistically significant (*p* > 0.05) (Table 4).

## 5. Discussion

XFS is the most important identifiable risk factor for open-angle glaucoma, although the mechanism responsible is still unclear [2]. The RNFL thickness decreases in diseases characterized by loss of ganglion cells, such as glaucoma. RNFL thickness measurement is useful in the early diagnosis and follow-up of glaucoma [14]. A decrease in RNFL thickness before the onset of visual field defect can prevent the progression of glaucoma by permitting early diagnosis and treatment [15]. Studies of eyes with XFS have shown that PEX material can cause RNFL loss.

Vergados et al. determined a lower RNFL thickness in a group with XFS compared to a control group [16]. Yu et al. reported a significant decrease in RNFL thickness in a group with XFS compared to a control group. Those authors also noted the importance of evaluating RNFL thickness using OCT in patients with XFS in order to identify early glaucomatous damage and initiate treatment before the onset of visual field loss [17]. Rao et al. reported statistically significantly lower RNFL thickness in patients with bilateral XFS compared to patients with unilateral XFS [5]. Özmen et al. also observed a thinner RNFL in an XFS group compared to a control group, although a significant difference was found only in the inferior quadrant [18]. Puska et al. suggested that PEX material may be a risk factor, independently of IOP, for ganglion cell loss in XFS and XFG [19]. The findings from the present study support that thesis. Mohammed et al. and Yüksel et al. compared eyes with XFS with normal contralateral eyes and a healthy control group, and similarly reported a significantly thinner RNFL in all quadrants in eyes with XFS, apart from in the nasal quadrant [5,20]. In the present study, RNFL thickness was lower compared to those in the other groups in all quadrants and on average. Significant temporal, inferiotemporal, and mean RNFL thinning was observed in the XFS group compared to the control group. A decrease in RNFL thickness in patients with XFS may be a sign that exfoliative material is by itself a risk factor and results in glaucomatous damage. For that reason, RNFL measurement using OCT is important in the detection of glaucomatous damage in eyes with XFS.

Elastin is one of the major components of the extracellular matrix of arterioles, and XFS resulting from this is regarded as a systemic vasculopathy. [1] Exfoliative material affects the anterior ocular segment but has also been shown to be capable of affecting the posterior segment, such as the ciliary arteries, the vortex veins, and the central retinal veins [6]. In addition, studies have also reported that it causes a decrease in ocular blood flow [7]. Galassi et al. investigated retrobulbar hemodynamics and ocular perfusion pressure and observed impaired retrobulbar hemodynamics and lower ocular perfusion pressure in patients with XFG compared to healthy individuals [21]. Sarrafpour et al. compared eyes with XFS and XFG with a control group and found that choroidal vessel diameters were smaller in affected eyes, and also reported that this change was independent of glaucoma [22]. Martinez et al. and Yüksel et al. reported a decrease in ophthalmic artery, central retinal artery, and short posterior ciliary artery blood flow, and an increase in vascular resistance in patients with XFG and XFS, as well as a significant change in retrobulbar hemodynamics [7,23].

The start of in vivo examination of the choroid and increasing importance of the choroid also led to greater examination of this highly vascularized layer. Due to its unmatched ability to provide high-contrast, depth-resolved retinal layer morphology in vivo with micrometer resolution, OCT has emerged as the gold standard for clinical retinal diagnosis over the last decade [24]. Göktaş et al. compared patients with XFS and a control group and reported significant choroidal thinning in the subfoveal, nasal, and temporal regions. Those authors suggested that this thinning might be related to a decrease in ocular blood flow and increased vascular resistance [25]. Bayhan et al. determined thinning in the nasal choroid in patients with XFG compared to a control group but observed no significant difference in the subfoveal and temporal regions [26]. Turan-Vural et al. reported significant thinning in the choroid and a decrease in ocular perfusion pressure in eyes with PEX material compared to a control group. Those authors suggested that, in addition to affecting the eye in clinical terms, PEX material also affected choroidal blood flow [27]. Zengin et al. compared patients with XFS and a control group and observed lower mean choroidal thickness in the XFS group, although the difference was not statistically significant [28]. Similarly, Moghimi et al. detected no significant change in macular choroidal thickness in patients with XFS compared to a control group [29]. Demircan et al. showed thinning in the choroid in patients with XFG and XFS and emphasized that accumulating exfoliative material weakened choroidal blood flow [30]. Özge et al. examined three groups, XFG, XFS, and control, in terms of choroidal thickness but observed no significant difference. Those authors concluded that a stable choroidal thickness and decrease in RNFL thickness, and the presence of PEX material, were a more significant risk factor than choroidal changes in the progression of XFS to XFG [31]. In the present study, macular choroidal thickness was significantly lower in individuals with XFG compared to individuals with XFS and the healthy controls. No significant difference was observed between the XFS and control groups.

To the best of our knowledge, there are no studies showing the extent to which length of exposure to exfoliation material affects progression. The lower choroidal and RNFL thicknesses in the 6–10 years disease duration group patients in the XFG and XFS groups in the present study compared to the 1–5 years duration group shows that the eye is affected by the length of pseudoexfoliation, that vascular resistance perfusion is impaired, that the thickness of the tissues in the posterior segment changes, and that this plays a role in the development of glaucoma.

The principal limitations of this study are the fact that choroidal thickness was measured manually and the antiglaucoma agents used by the XFS patients may also have affected choroidal thickness. While some studies in the literature have reported that antiglaucoma agents affect choroidal thickness, others have observed no such effect [32,33].

In conclusion, decreases in both choroidal and RNFL thicknesses in patients with XFG and XFS shows that PEX material may be a risk factor for the progression of XFS to XFG. In addition, in terms of length of exposure to PEX, the lower choroidal and RNFL thicknesses in the 6–10 years disease duration patient group show the importance of duration of exposure to PEX material. We conclude that greater exposure to PEX material of the neuronal and vascular tissues of the eye may also exacerbate its deleterious effects. Our detection of thinning in the RNFL in eyes with XFS compared to normal eyes shows that glaucomatous changes begin before the diagnosis of glaucoma in XFS. Regular OCT measurements of eyes with PEX are important in terms of early detection of potential glaucoma and of early treatment. The present study now needs to be supported by further research with larger patient groups in order to clarify the effect of PEX material on choroidal circulation, the association with choroidal thickness, and the relationship between decreased RNFL thickness and the development glaucomatous damage.

## Figures and Tables

**Figure 1 diagnostics-13-00314-f001:**
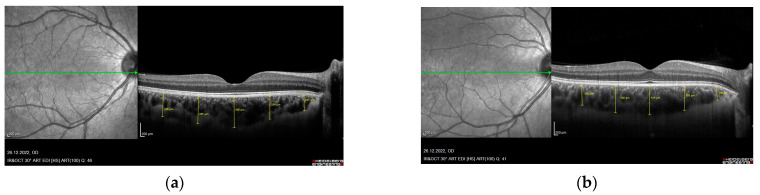
Optical coherence tomography image showing macular choroidal thicknesses in five different locations. (**a**) XFG, (**b**) XFS.

**Figure 2 diagnostics-13-00314-f002:**
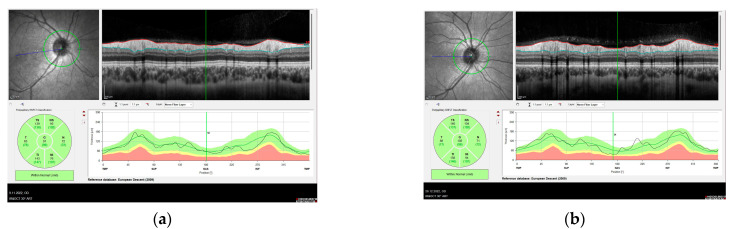
Optical coherence tomography image showing peripapillary RNFL thickness. (**a**) XFG, (**b**) XFS.

**Table 1 diagnostics-13-00314-t001:** Demographic and ocular characteristics of the groups.

Variable	XFG	XFS	Control	*p*
Age (years)	56.51 ± 5.37	56.48 ± 6.82	55.31 ± 7.55	0.508
Sex (female/male)	8/15	7/15	8/15	0.879
IOP (mmHg)	17.62 ± 1.78	17.02 ± 1.77	16.34 ± 2.80	0.612
Axial length (mm)	22.68 ± 0.94	22.90 ± 0.96	22.81 ± 0.73	0.523
CCT (μm)	524.73 ± 7.12	525.61 ± 6.17	525.44 ± 2.96	0.658
MD (dB)	−5.28 ± 2.40	−1.48 ± 0.64	−0.54 ± 0.30	<0.001

MD = Mean deviation, CCT = central corneal thickness.

**Table 2 diagnostics-13-00314-t002:** RNFL thicknesses in eyes with XFG and XFS and healthy control eyes (µm).

	XFG	XFS	Control	*p1*	*p2*	*p3*
Temporal	57.94 ± 14.95	68.05 ± 9.91	74.67 ± 13.50	<0.001	<0.001	<0.001
Superotemporal	89.11 ± 35.39	134.32 ± 14.35	137.13 ± 20.60	<0.001	<0.001	0.419
Superonasal	73.50 ± 28.41	104.56 ± 21.77	112.66 ± 17.24	<0.001	<0.001	0.006
Nasal	57.51 ± 16.56	76.33 ± 12.45	84.68 ± 14.06	<0.001	<0.001	0.001
Inferonasal	76.82 ± 27.71	117.82 ± 23.29	125.63 ± 17.52	<0.001	<0.001	0.019
Inferotemporal	89.62 ± 34.62	137.00 ± 25.98	149.59 ± 19.23	<0.001	<0.001	<0.001
Mean	69.53 ± 19.50	97.47 ± 6.71	104.20 ± 7.91	<0.001	<0.001	<0.001

RNFL: Retinal nerve fiber layer thickness (µm). P1: Comparison of the XFG and XFS groups. P2: Comparison of the XFG and control groups. P3: Comparison of the XFS and control groups.

**Table 3 diagnostics-13-00314-t003:** Macular choroidal thicknesses in eyes with XFG and XFS and healthy control eyes (µm).

	XFG	XFS	Control	*p1*	*p2*	*p3*
Subfoveal	270.42 ± 66.33	326.65 ± 42.88	335.80 ± 51.88	<0.001	<0.001	0.154
1.5 mm nasal	207.03 ± 68.40	231.08 ± 55.97	260.71 ± 52.42	<0.001	<0.001	0.001
3.0 mm nasal	145.78 ± 41.12	162.42 ± 33.53	174.80 ± 45.39	<0.001	<0.001	0.145
1.5 mm temporal	226.00 ± 61.37	274.52 ± 41.85	287.66 ± 50.60	<0.001	<0.001	0.206
3.0 mm temporal	194.69 ± 52.30	217.16 ± 35.30	217.40 ± 41.58	<0.001	<0.001	0.925
Mean	208.78 ± 51.91	242.37 ± 34.79	244.66 ± 55.24	<0.001	<0.001	0.496

P1: Comparison of the XFG and XFS groups. P2: Comparison of the XFG and control groups. P3: Comparison of the XFS and control groups.

**Table 4 diagnostics-13-00314-t004:** The effect on choroidal and RNFL thickness of 1–5 and 6–10 years disease duration in the XFG and XFS groups.

	1–5 Years	6–10 Years	*p*
Mean choroid in XFG	213.96 ± 54.21	203.60 ± 49.65	0.412
Mean choroid in XFS	243.82 ± 38.28	240.91 ± 31.35	0.787
Mean RNFL in XFG	70.87 ± 20.44	68.20 ± 18.69	0.519
Mean RNFL in XFS	97.61 ± 5.62	97.33 ± 7.73	0.443

RNFL: Retinal nerve fiver layer thickness (µm).

## Data Availability

The data that support the findings of this study are available from the corresponding author, upon reasonable request.

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
