# Peer review of "Evaluation of the Effect of Duration on Retinal Nerve Fiber Layer Thickness and Choroidal Thickness in Exfoliation Syndrome and Exfoliative Glaucoma"

_diagnostics, 2023, doi:10.3390/diagnostics13020314_

Round 1

Reviewer 1 Report

Authors have used SD-OCT retinal images of the patients with exfoliation syndrome (XFS) and exfoliative glaucoma (XFG) to assess the thickness of RNFL and choroid relative to healthy controls. They reports that assessment has been performed on two age groups. They reports that RNFL thickness  was statistically significantly lower in eyes with XFG and XFS than in the control group. Macular choroidal thickness decreased significantly in the XFG group compared to the XFS and control groups. No significant difference was observed between the XFS and control  groups. This is an interesting unique piece of clinical study. Overall, the manuscript is written well, albeit, need a revision based on the following minor comments.

1. I didn't fully understood the age group mentioned. Do the age group  1-5 Yr  and 6-10 year-old represent  the age of the patients  or disease ? Need to clearly mention.

2. Authors should provide the the segmented retinal OCT B-scan images and thickness map for both diseases and relative control for both 'age groups'.

3. Authors are advised to add a general comment on OCT technology to emphasize its potential for in vivo structural and functional imaging of retina non-invasively across different aerial, terrestrial and  aquatic verterbrate species [1-2] with the suggested references. (https://doi.org/10.1167/tvst.11.8.11 ; https://doi.org/10.1038/s41598-021-95320-z )

Author Response

Please see the attachment.Please see the attachment.

Reviewer 2 Report

It is a very interesting topic. Analysis of the nerve fiber layer and choroid thickness under these conditions has been previously studied, but it is true that there is no reference to the effect of the exfoliative material over time.

The paper is well developed but some aspects related to the sample, definition of the objective, results and explanation of the limitations should be improved.

ABSTRACT:

Line 5: In the abstract you say that the objective of your study is to compare RNFL thickness and choroidal thickness in different groups of patients, right? As the following paragraph is written, (“To compare retinal nerve fiber layer (RNFL) thickness and choroidal thickness in 5 1-5 and 6-10-year-old patients with exfoliation syndrome (XFS) and exfoliative glaucoma (XFG) 6 compared to healthy volunteers”). It seems that there are two groups of children of different ages, one group of children between 1 and 5 years and other group of children between 6-10 years old? But in the introduction in line 79 you say that Individuals aged under 18, were excluded from the study and the table 1 show results for adults.

I understand that you want to indicate that the thickness measurements will be evaluated after a period of between 1 and 6 years or between 5 and 10 years.

It makes more sense to have included adult patients than children as young as 1 year of age.

I don't know how some of the protocol tests could be performed on such young children, for example, it is considered almost impossible to perform a visual field  on 1-year-old children.

Please clarify this point because it is very important. Explain in detail, how the length of exposure to disease was measured. (line 151)

MATERIAL AND METHODS

Line 53: Why you decided to include 78, 78 and 83 patients in each group? You have said that one of the limitations of this study is the low case number. Have you done a sample size calculation?

Line 66: how have you done the classification of XFS and XFG groups? Have you used previous protocol? For example,  Aghsaei Fard M et al in their pape about choroidal microvasculature in  Pseudoexfoliation Syndrome and Pseudoexfoliation Glaucoma define PXS group such as patients that had visible pseudoexfoliation material on the anterior lens capsule or pupillary margin after mydriasis on slit- lamp, an intraoperative pressure (IOP) <22 mm Hg with no history of increased IOP, an absence of glaucomatous disc appearance, and a normal visual field defined by mean deviation and pattern standard deviation within 95% confidence interval limits and a Glaucoma Hemifield Test within normal limits.

You must define what is a normal visual field finding.

Line 84: In addition, patients using 84 prostaglandin group drugs were excluded from the study. Explain the reason to exclude prostaglandin agents instead of others. Has it shown that prostaglandin may affect choroidal thickness? In this case, explain it.

Line 97: How can you ensure that the measurement of the choroid when doing it manually is reliable? How can you know if the measurement over time is always made at the same point?

RESULTS

You can include some graphs showing retinal nerve fiber layer (RNFL) thickness profiles such in the following paper: Retinal Nerve Fiber Layer and Peripapillary Choroidal Thicknesses in Non-Glaucomatous Unilateral Optic Atrophy Compared with Unilateral Advanced Pseudoexfoliative Glaucoma
